# Phenotypic Diversity and Genetic Parameters of *Coffea canephora* Clones

**DOI:** 10.3390/plants12234052

**Published:** 2023-12-01

**Authors:** Caroline de Souza Bezerra, Jennifer Souza Tomaz, Mágno Sávio Ferreira Valente, Marcelo Curitiba Espindula, Ricardo Lívio Santos Marques, Hugo Cesar Tadeu, Fábio Medeiros Ferreira, Gabriel de Sousa Silva, Carlos Henrique Salvino Gadelha Meneses, Maria Teresa Gomes Lopes

**Affiliations:** 1Faculdade de Ciências Agrárias, Universidade Federal do Amazonas, Avenida Rodrigo Otávio Ramos, 3.000, Bairro Coroado, Manaus 69077-000, AM, Brazil; caroline.bezerra@ufam.edu.br (C.d.S.B.); monitorlabgenvegetal@ufam.edu.br (J.S.T.); liviorlsm@ufam.edu.br (R.L.S.M.); hugocesartadeu@ufam.edu.br (H.C.T.); ferreirafmt@ufam.edu.br (F.M.F.); 2Instituto Federal do Amazonas, Figueiredo. Av. da Onça-Pintada, S/N—Galo da Serra, Presidente Figueiredo 69735-000, AM, Brazil; magno.valente@ifam.edu.br; 3Centro de Pesquisa Agroflorestal de Rondônia, Empresa Brasileira de Pesquisa Agropecuária, Rodovia BR-364, Km 5,5, Porto Velho 76815-800, RO, Brazil; marcelo.espindula@embrapa.br; 4Programa de Pós-Graduação em Ciências Agrárias, Departamento de Biologia, Centro de Ciências Biológicas e da Saúde, Universidade Estadual da Paraíba, Campina Grande 58429-500, PB, Brazil; gabriel.sousa.silva@aluno.uepb.edu.br (G.d.S.S.); carlos.meneses@servidor.uepb.edu.br (C.H.S.G.M.)

**Keywords:** plant genetic breeding, intervarietal hybrids, multivariate analysis, heritability

## Abstract

The simultaneous analysis of the maximum number of chemical elements present in plant tissues provides more comprehensive information about their chemical constitution and increases the number of characteristics for the selection process in various plant breeding programs. The objective of this study was to analyze productivity, grain yield, and concentration of chemical elements in tissues of *Coffea canephora* clones to study phenotypic diversity and estimate genetic parameters for use in breeding. This experiment was carried out in Manaus, Amazonas, Brazil, in randomized blocks with four replications. The concentrations of elements in various organs were quantified using total reflection X-ray fluorescence (TXRF). Genetic parameters and genetic divergence were estimated, and genotypes were clustered using the UPGMA hierarchical method and non-metric multidimensional scaling analysis. The study allowed us to differentiate the performance of the clones in terms of the absorption of essential and non-essential chemical elements for plant development and to analyze the correlation of the characteristics in the selection process. TXRF efficiently characterizes the presence and concentration of multiple elements, aiding genotype discrimination for *C. canephora* improvement.

## 1. Introduction

Coffee is a highly commercialized and widely consumed primary product worldwide [1,2]. There are two species of the Coffea genus that have significant socioeconomic importance: *Coffea arabica* L. and *Coffea canephora* Pierre ex A. Froehner. The production of *C. canephora* coffee has been increasing worldwide. Between October 2021 and September 2022, considering the volume of a 60 kg bag, 170.83 million and 164.9 million were produced and consumed globally, respectively. Brazil is the world’s largest producer and exporter of coffee, accounting for approximately 35% of global production [3,4,5,6]. The species, adapted to tropical regions, has seen its clonal cultivation expand in the Northern Brazilian region. This has become a significant and lucrative agricultural activity benefitting the states of Amazonas and Rondônia, as well as states in other regions such as Espírito Santo (Southeast) and Bahia (Northeast) [6,7,8].

It is an allogamous species, of easy vegetative propagation and presents a self-incompatibility system of the gametophytic type [6,7]. It has high vegetative vigor, high rusticity, and productivity [5,9]. Recombination within the species generates highly heterozygous genotypes that exhibit diverse adaptation and acclimatization skills, as well as varied behaviors related to nutrient absorption and response [6,10,11,12].

Balanced nutrient management is necessary due to genetic variability in *C. canephora* plantations, ensuring high yields can be obtained and maintained in each production cycle [6]. According to previous studies, genetic factors may affect nutrient availability tolerance, uptake efficiency [13], content accumulation in leaves and branches [11], and beverage quality [14]. The genetic variability as the absorption of chemical elements can be analyzed in controlled experiments [15,16]. It is possible to use multielemental analyses and estimate genetic parameters to select superior genotypes based on phenotypic diversity.

The availability and presence of chemical elements in the soil have a significant impact on the absorption and accumulation of plants. By monitoring the accumulation of these elements in different plant tissues, we can determine the effectiveness of different genotypes in plant breeding and agriculture. [17]. The use of total reflection X-ray fluorescence spectrometry (TXRF) is considered advantageous and has been widely used to quantify multielement concentrations in rhizosphere soils and plant tissues [18]. This is a technique that presents high reproducibility of results, quick measurement, and the ability to detect many elements of the periodic table from a single sample. This method requires comparatively smaller quantities of material for analysis, and reading is simplified using the internal standard method, which involves adding a known amount of an element [18,19,20].

TXRF detects not only the presence of chemical elements essential to the plant and human health but also the harmful elements and their concentration. Plant breeding has become increasingly specialized to produce more productive cultivars that offer higher-quality products. It is necessary to estimate genetic parameters to select genotypes that have high absorption efficiency of essential elements and low absorption efficiency of harmful elements, which can be toxic at high concentrations. The presence of trace elements in soils is a common occurrence, but it is not always monitored via chemical analysis in crop production areas. It is necessary to expand information on all chemical elements absorbed by plants due to the increased contamination of environments in modern society and the demand for safer products.

This study presents an unprecedented application of the TXRF technique to analyze the multielement content of various organs of *C. canephora* in a controlled experiment. The results allowed us to estimate genetic parameters such as heritability and character correlations for the selection process and to obtain information on the efficiency of using multielemental analysis via TXRF as an evaluation tool for use in plant breeding.

The application of the TXRF technique in the tissue analysis of different plant organs can be investigated for efficiency in the absorption of chemical elements. The research hypothesis was to verify if the results of the estimation of genomic parameters, such as variances and heritability, using the TXRF analysis are helpful in the discrimination of genotypes in coffee breeding. The TXRF analysis performed in a controlled experiment of *C. canephora* in this work is unprecedented.

We evaluated the grain yield and productivity of *C. canephora* clones. Additionally, the concentrations of chemical elements simultaneously present in plant tissues (root, leaf, raw, and roasted grains) were analyzed using TXRF. The research objective was to determine the phenotypic diversity for chemical elements, productivity, and grain yield and obtain the genetic parameters to characterize the clones for plant breeding.

## 2. Results

### 2.1. Genetic Parameters and Comparison of the Mean Concentration of Chemical Elements in Leaves, Grain Yield, and Productivity

The coefficient of environmental variation (CVe) for the different chemical elements in leaf analysis ranged from 5.32% to 181.11%, as shown in Table 1. The CVe values for yield and productivity were 14.30% and 25.95%, respectively. The highest values of CVe were found for the following trace elements: I (181.11%), Rb (52.19%), Cu (45.95%), Ni (40.54%), Ti (44.04%), and Cr (37.88%). Low values of CVe were found for the following elements: K (5.32%), P (6.01%), Cl (9.30%), Zn (13.43%), S (14%), and Ca (16.11%). In general, the values of CVe for the leaf analysis of elements introduced via soil fertilization, such as K, P, Zn, S, and Ca, were lower than those not introduced into the soil; this included I, Rb, Cu, Ni, Ti, and Cr. This observation is depicted in Appendix A.

The range of coefficients of genetic variation (CVg) for chemical elements in leaf analysis was 1.26% to 76.37%, as shown in Table 1. CVg values for grain yield and productivity were 10.35% and 43.75%, respectively. The trace elements Hf, I, and Ti had the highest CVg values (76.37%, 50.34%, and 38.39%, respectively), while the elements K, P, Cu, Cr, and Zn had the lowest CVg values (1.26%, 1.74%, 3.18%, 4%, and 8.87%, respectively).

The ratio of CVg/CVe for leaf analysis varied from 0.07% to 1.85%, and the heritability ranged from 1.88% to 93.19%. The values of CVg/CVe for grain yield and productivity were 0.72 and 1.69, respectively. Productivity (91.91%) had a higher heritability than grain yield (67.69%). CVg/CVe ratios for leaf analysis higher than 1 were found for Br (1.17%) and Hf (1.85). The highest H^2^ values were also found for Br (84.50%) and Hf (93.19%).

The analysis of variance showed significance for the effect of treatments for productivity, grain yields, and chemical elements Ti, Fe, Zn, Br, Rb, Sr, Y, and Hf (Table 1). Additionally, the Scott–Knott test revealed significant statistical differences in the mean concentrations of chemical elements in *C. canephora* clones’ leaves for Co, Cu, Rb, and Hf (Appendix A). Clone BRS2314 presented the highest average (12.07 mg·kg^–1^) for Co absorption. Clone BRS3220 and Clone 15 presented the highest averages, 3703.05 mg·kg^–1^ and 1966.91 mg·kg^–1^, respectively, for Cu absorption. The clones that absorbed the most Rb were BRS3220 (6438.32 mg·kg^−1^), BRS2299 (6102.0 mg·kg^−1^), RO_C125 (5984.23 mg·kg^−1^), BRS2336 (5534.25 mg·kg^−1^), BRS2357 (5161.86 mg·kg^−1^), and RO_C160 (4673.47 mg·kg^−1^). The clones with the highest absorption averages for Hf were RO_C125 (4881.25 mg·kg^−1^), BRS3220 (4666.50 mg·kg^−1^), BRS2336 (3959.25 mg·kg^−1^), BRS2357 (3068.22 mg·kg^−1^), RO_C160 (3061.25 mg·kg^−1^), Clone 15 (2826.27 mg·kg^−1^), Clone 9 (2767.00 mg·kg^−1^), and BRS3193 (2701.50 mg·kg^−1^). For grain yield, considering a 60 kg bag, the means of clones BRS2314 (0.303), BRS2357 (0.300), RO_C160 (0.300), BRS1216 (0.288), and Clone 9 (0.270) (Appendix A) were higher than those of others and the general average of the experiment (0.26) (Table 1). Three clones with superior grain yield, BRS1216 (116.53), RO_C160 (111.71), and BRS2357 (102.12), also had higher productivity, in bags of 60 kg·ha^−1^ (Appendix A).

### 2.2. Multivariate Analysis for Concentrations of Chemical Elements of Roots, Leaves, Raw and Roasted Grains, Grain Yield, and Productivity of Clones

#### 2.2.1. Grouping of Genotypes Based on the Concentration of Chemical Elements in the Roots

The clones of *C. canephora* were divided into three groups (Figure 1): Group 1 consisted of Clone BRS3213, Group 2 included BRS2336, and Group 3 comprised 13 clones. Group 3 was divided into two subgroups. One subgroup had four clones (Clone 15, BRS3220, BRS2299, and BRS3193). The other subgroup had the remaining nine clones (BRS3137, Clone 9, BRS3220, RO_C125, RO_C160, BRS2314, BRS2357, BRS1216, and Clone 12).

The representation of the similarity relationship between the root samples of two-axis clones was performed using non-metric multidimensional scaling analysis and confirms the separation of genotypes into the three groups (Figure 2).

#### 2.2.2. Grouping of Genotypes Based on the Concentration of Chemical Elements in the Leaves

According to the hierarchical method UPMGA, which uses the Euclidean distance as a measure of dissimilarity, the concentration of chemical elements in the leaves formed two distinct groups. The analysis revealed that one group included only a single clone, identified as BRS2314, while the other larger group comprised the remaining 14 clones (as shown in Figure 3).

The representation of the similarity relationship between the samples of *C. canephora* clone leaves from two axes was performed using the non-metric multidimensional scaling analysis (Figure 4). The division of genotypes into two distant groups was confirmed.

According to the Singh method [22], the concentrations of certain chemical elements in leaves play a significant role in discriminating genetic diversity. Hf (16.35%), Rb (14.09%), Co (12.03%), and I (8.01%) together contributed to 50.48% of the discrimination, whereas the other elements had a contribution of 7% or less (Table 2). It is worth noting that the macronutrient K also showed a good contribution (6.61%).

#### 2.2.3. Grouping of Genotypes Based on the Concentration of Chemical Elements in Raw and Roasted Grains

The UPGMA revealed two groups for the concentration of chemical elements in the raw grains: Group 1 included Clone 9, and Group 2 comprised the remaining 14 clones (Figure 5a). The genotypes were classified into two major groups for the roasted grain (Figure 5b). Group 1 included eight clones: BRS2314, BRS2336, Clone 12, BRS2357, RO_C125, RO_C160, Clone 9, and Clone 15. Group 2 had seven clones: BRS3137, BRS3213, BRS1216, BRS3210, BRS3220, BRS2299, and BRS3193.

The NMDS analysis confirmed the division of raw grains into two groups, as shown in Figure 6. Moreover, it is possible to observe a higher dissimilarity for Clones BRS1216, BRS3210, and BRS2336, as well as Clone 9, which was placed in a separate group (Figure 6a). Two groups were confirmed for roasted grains. Group 2 showed greater dispersion among clones than Group 1 (Figure 6b). The dissimilarity in roasted grains between Clones BRS2336 and BRS3193 was the greatest.

#### 2.2.4. Grouping of Genotypes Based on Grain Yield and Productivity

Two large groups were formed using the UPGMA for grain yield and productivity, one with seven clones and the other with eight. Group 1 includes the following genotypes: BRS1216, BRS2336, BRS2299, BRS2357, RO_C160, Clone 9, and Clone 15. Group 2 consisted of BRS3137, BRS3213, BRS2314, BRS3210, BRS3220, BRS3193, Clone 12, and RO_C125 (Figure 7).

The NMDS analysis confirmed the separation of the two groups. However, Clones BRS3137 and BRS2336 were located at a greater distance from their respective groups (Figure 8). It is possible to observe a greater dispersion among the clones of Group 2 than among Group 1.

### 2.3. Correlation between the Productivity of C. canephora and Chemical Element Concentration in Leaves and Raw and Roasted Grains 

Using the Pearson correlation coefficient for concentrations of chemical elements in leaves, raw and roasted grains, and the biomass (dry mass of the leaves and roots, total dry mass, leaf mass ratio, and root mass ratio) of 15 clones of *C. canephora*, a total of 113 significant correlations were detected at 1% and 5% probability using a *t*-test, comprising 88 positive and 25 negative correlations (Figure 9).

High values of positive correlations between chemical element concentrations in the leaves were observed for Co × Ni (1.0), Fe × Ni (0.964), Fe × Co (0.964), Cr × Fe (0.834), Cr × Ni (0.819), and Cr × Co (0.812) (Figure 9). For raw grains, high positive correlations were identified between the elements Br × Sr (0.981), K × Rb (0.919), Cr × Ni (0.913), Cu × Rb (0.830), and Ti × Co (0.802). Positive correlations were observed between the concentrations of the chemical elements in roasted grains ( close to 1.0) for Ti × Rb, Ti × Fe, Fe × Rb, Cr × Fe, Fe × Ni, Ti × Cr, Ti × Rb, Ti × Ni, Ni × Cu, Ni × Rb, Cr × Ni, Fe × Cu, Ti × Cu, and Tr × Cu. High positive correlation values were observed between the leaf dry mass and total dry mass (0.843), as well as between the Hf of roasted grains with Ni (0.870) and Cr (0.809). Additionally, there was a strong positive correlation between productivity and grain yield (0.791). Negative correlations with a modulus equal to or greater than −0.8 were found between the K of roasted grains and Cr, Cu, Ni, Fe, Ti, and Rb.

## 3. Discussion

### 3.1. Genetic Parameters and Comparison of the Mean Concentration of Chemical Elements in Leaves, Grain Yield, and Productivity

The highest values of CVe detected for the concentrations of elements in the leaves, in general, were for trace elements not introduced into the soil.

High values of CVe are associated with crop cycle, experiment size, crop management, genotype responses to high temperatures, drought, and the incidence of pests and diseases [23,24]. Some of the methods for improving the accuracy of the detection of these elements involve performing controlled experiments in a greenhouse, in pots with substrate standardization, or through in vitro cultivation. There is no classification in the literature for the magnitude of CVe and CVg values specific to trace elements for the characteristics evaluated in field trials for *C. canephora*.

The values of CVe for the character yield and productivity were found to be nearly identical to those found in the literature for coffee, which demonstrates a good level of experimental accuracy. In Rondônia, Brazil, a state with climatic conditions like those in the Amazon, the clones of *C. canephora* had CVe values of 31.20% [25] and 35.87% [26] for grain yield. In Planaltina, Central Brazil, the progenies of *C. canephora* were evaluated, and CVe values of 12.048% for grain yield and 11.578% for productivity were found [27].

A CVg/CVe ratio greater than or equal to 1.0 and higher heritability values in the broader sense indicate less influence of the environment in the control of the characteristics and more favorable conditions in the genotypic selection of the characteristics [10,24,28]. It is noteworthy that high heritability was also obtained for productivity in a previous study on sixteen genotypes evaluated in Rondônia, Brazil [15] and for the chemical elements of leaves [6]. It is emphasized that the heritability for productivity, in general, presents the lowest values compared to other characteristics, especially when considering the joint analysis of environments.

For the concentration of the elements Br and Hf in leaves and productivity, it is possible to discriminate genotypes safely for clonal reproduction due to the low influence of the factors that most hinder the recognition in selection, the effects of the environment, and dominance. The high CVg/CVe ratio and heritability for these traits prevent undesirable genotypes from being selected due to high phenotypic values for the benefit of environmental action. Furthermore, proven superior genotypes are discarded because their performance is impaired by the environmental effect. Complete dominance prevents the distinction of heterozygous-dominant homozygotes but is eliminated in clonal propagation after the identification of the superior individual in genotypic selection [6,29,30].

The selection of trace elements related to the different stages of coffee tree development, such as in the leaves, must be taken into consideration in plant-breeding programs to establish a correlation with their impact on productivity, grain yield, and the quality of the beans.

The grain yield and productivity of *C. canephora* are the main characteristics used in selection for the clonal multiplication of plants [31]. Although productivity is one of the most critical criteria used in the selection of plant breeding [26,32], in this study, it was found to be an appropriate characteristic for genotype discrimination.

### 3.2. Multivariate Analysis of Grain Yield, the Productivity of Clones, and Concentrations of Chemical Elements in Roots, Leaves, and Raw and Roasted Grains 

The division found in the numbers of groups and composition of genotypes by plant tissue shows that the concentration of the elements behaves differently depending on the tissue used in the analysis.

By obtaining divergent groups of species, clones, heterotic groups, and potential parents can be identified to determine crosses for characteristics of interest [16,17,29,33]. Genetic distance studies in *C. canephora* are essential for the selection of parents in planning strategies and advances in breeding programs. 

The formation of genetically distant groups for the characteristics studied confirms the potential use of genetic variability available for use in breeding and selection. Six groups have already been observed for the concentrations of chemical elements in the different plant tissues, leaves, flowers, and grains of 16 Robusta coffee genotypes, three of which contained only one genotype [6]. A similar result was obtained for the genetic diversity of chemical elements in the grains of 20 genotypes of *C. canephora*, in which, among the six formed groups in the detected cluster, three groups comprised a single genotype [16].

Considering the use of genotypes in the present study that remained in the same group, these may also be useful for obtaining cultivars with homogeneity for the characteristics studied and may be of benefit to the agricultural industry in order to facilitate the management of the crop based on the affinity in the uptake and accumulation of certain chemical elements by the various plant tissues, grain yield, and productivity of *C. canephora* [15,34,35].

To increase the reliability of dissimilarity between the genotypes grouped for the characteristics investigated in this study, the NMDS analyses that were performed confirmed the separation of clones performed with the UPGMA. Using methods, namely the UPGMA and NMDS, that analyze the heterogeneity of genotypes is essential for confirming the results, ensuring the reliability of genetic distances, and increasing the power of the analysis to differentiate genotypes [6,35,36,37]. The cluster analysis did not always verify the grouping of the half-brothers. The clones’ kinship degree, considering the origin of the group of brothers “Encapa” is only by one of the parents. Half-brothers share only half of their genetic material. Characteristics that are not common to genotypes and not determined by the common parent can often be determinants in the cluster analysis.

The results regarding the contribution to genetic diversity show the most relevant chemical elements for the analysis and those that can be excluded for presenting low values [16,38]. The presence of natural trace elements (i.e., not intentionally introduced into the soil) that are in high concentrations and can be used for genotype discrimination helps to form a hypothesis regarding the selection of plants that absorb and accumulate these elements.

### 3.3. Concentration of Chemical Elements in Soil and Tissues (Roots, Leaves, and Raw and Roasted Grains) of C. canephora

Selection in coffee breeding has concentrated on obtaining genotypes with higher productivity and grain yield, among other characteristics such as uniform maturation, grain size, and tolerance to biotic stresses [32]. In terms of the relevance of the “coffee culture”, coffee is considered the most consumed beverage in the world after water. Breeding programs need to advance in selecting and improving grain quality characteristics. The selection of trace elements essential to plants and necessary for human or animal health and selection against natural elements that are toxic in high concentrations may also result in a better-quality drink with higher market value.

The Al and Si found in rhizospheric soils were expected because they are elements found in high concentrations in Amazonian soils and abundant in the Earth’s crust [39,40]. Plants with the genotypes of *C. canephora* can still be considered accumulating plants since Al and Si concentrations of greater than 1000 mg·kg^−1^ can accumulate in their roots [41]. In addition, the complexation of aluminum (Al) in the roots, avoiding its transport to the aerial parts of the plants, characterizes these clones as tolerant [42].

Ba was detected only in rhizospheric soils and in roasted grains. According to Martinez et al. [14], the roasting process comprises steps that alter the chemical and physical composition of the grains, which can contribute to the presence of these elements considered unusual. In addition, Ba can be deposited on surfaces through emission from vehicles such as tractors [43].

Regarding the elements K, Ca, S, P, Cl, Fe, Ti, I, Zn, Cr, Sr, Hf, and Y, the effect of the concentration values of these elements is not precisely known, nor is their interaction with the plant or their impact on human and animal health. Among the macronutrients, the elements Ca, K, P, and S were found at higher concentrations in the roots and in the raw and roasted grains of *C. canephora*. Among the micronutrients, Cl and Zn were detected at higher concentrations in roots and Fe in rhizospheric soils, roots, and roasted grains of the clones.

The high concentrations of Ca and K found in the roots and grains of the clones have been previously reported for *C. canephora* due to a higher demand for the grain-filling period [6,11,44,45]. Other elements found in high concentrations in the fruit formation phase were S and P; these are essential for grain regulation and quality [45,46,47,48].

The concentrations of Cl and Zn were deficient in rhizospheric soils; however, they were high in roots. As for Cl, despite being a nutrient that assists in plant metabolism, data on its presence in coffee trees were not found. Zn is one of the most essential elements for the species, mainly in acid soils such as the Amazon [49], and its high concentration in coffee roots has been observed in several studies [44,49].

Some studies also showed high levels of Fe in roots [44,50,51] and coffee fruits [11,45,51,52,53]. However, no statistical difference was observed in the absorption or accumulation of this element in the leaves. Still, it is one of the most absorbed and accumulated elements in conilon coffee plants [34].

The Ti, I, and Cr found are considered trace elements. Data defining the reference values for the accumulation of these elements in plant tissues and organs are not available in the literature. Ti, I, and Cr have environmental benefits and are essential for animal and human nutrition. Notwithstanding, they can be toxic to plants depending on their concentration [54].

The roots accumulated more Ti, and a smaller amount was transported to the leaves and grains. This element is cited as a plant growth promoter and redox catalyst [55]. The presence of I in grains is beneficial, although it is necessary to define its adequate amounts in future studies. Iodine has greater relevance for humans and animals, but it also plays a role in plant metabolism [56] and has been studied as a biofortifier in coffee [57]. The presence of a more significant concentration of I would be a solution to increase antioxidant defense in humans with I deficiency since coffee is one of the most consumed beverages in the world [58].

It is widely recognized that Y, found in rhizospheric soils, roots, and raw and roasted grains, is a rare earth element from the Amazon. The magnitude of its possible impacts on the ecosystem and human health is relatively unknown, especially in tropical systems [59]. Sr, a natural and commonly occurring alkaline earth metal [60], showed the exact behavior of Y in the clones studied. It should be noted that Sr can accumulate in greater quantity in the tissues of the aerial part of the plant [61]. Hf, along with Y and St, have not been reported in coffee tissues, and knowledge about their biogeochemistry is scarce, as they are more related to occurrence in soils [62].

Using elements such as Hf in the industry has grown steadily in recent years, so it is expected to be present in the environment [63,64]. This would justify the concentrations in rhizospheric soils and tissues of *C. canephora* clones. Similar behavior was observed for Rb, considered a chemical analog of P. It is assumed that if there is a higher concentration of Rb in the soil than P, plants will absorb it in more significant amounts, which can affect their development [62]. Cu was present in all plant tissues, but it has already been recognized as one of the elements present in different parts of *C. canephora* plants [53].

Plants of this species can absorb nutrients differently against a concentration gradient and accumulate them, as presented in the present work. Some genotypes were able to absorb or translocate more nutrients than others. Differences in nutritional concentrations between genotypes evaluated in the same period were also observed by Silva et al. [17], Martins et al. [31], and Gomes et al. [35]. According to these authors, the differences may be related to factors such as the affinity of nutrient absorption, compartmentalization in roots or other organs, mobility in the xylem and phloem vessels, and changes in the rhizosphere during growth. The nutritional requirement of species varies according to the genetic characteristics of the plant and the vegetable organ analyzed, as well as the edaphoclimatic conditions, the time of cultivation, the age of the plant, and the cycle of maturation of the genotype [11,46].

Among the elements relevant to plant nutrition, it is worth noting that nitrogen (N) was not detected. Elements with atomic numbers (z) below 13, such as N (Z = 7), are challenging to evaluate with conventional TXRF instruments. These elements are low energy in fluorescence emission and are quickly absorbed in the path, not resulting in reading in the equipment [65]. This is considered a limitation of the technique for analyses with a nutritional focus, but it was not the main objective of this research and therefore did not affect the results.

### 3.4. Correlation between the Nutritional Concentrations of Leaves and Raw and Roasted Grains

The results indicate that the concentrations of trace elements in plant tissues influence the accumulation of other essential elements for plant development, growth, and productivity. In the work of Schmidt et al. [6], positive correlations for all nutrients were observed between leaf samples and other plant tissues. Santos et al. [33] and Lana et al. [66] also showed positive and negative correlations between leaf samples and other plant tissues, with the majority being positive, as also observed in this study.

The high positive correlations found for non-essential elements, such as Ni × Ti, Ni × Cu, Ni × Rb, and Cr × Ni, facilitate the selection process against these elements since selecting only against Ni would already be selecting against Ti, Cu, Rb and Cr.

For roasted grains, a reduction in K, which was correlated with an increase in the elements Cr, Cu, Ni, Fe, Ti, and Rb, was observed. Research has shown that K leaching occurs due to higher ion content and electrical conductivity, which may be associated with high temperature [67]. Thus, the percolation of K may be associated with the roasting process of the grains. Several temperature analyses of the roasting process are necessary for the grains of clones to study the concentration of K since this is related to beverage quality.

## 4. Materials and Methods

### 4.1. Experimental Installation and Description of Area and Plant Material

The experiment was conducted at the Experimental Farm of the Federal University of Amazonas (FAEXP/UFAM), Amazonas, Brazil. It lies at latitude −2°64′96 S, longitude 60°05′25 W, at km 38 of Rodovia BR 174, Ramal, Presidente Figueiredo, AM, Brazil. The region has a tropical climate classified as Aw, according to the Köppen classification, and has two distinct seasons: a dry season that occurs between June and October (Amazon summer) and a rainy season between November and May (Amazon winter) [6]. The area’s soil is classified as a secondary forest [39].

Before planting, we analyzed a sample composed of soil from the site that presented pH (H_2_O) of 5.03; 289.10 g·kg^−1^ of organic matter; 178 mg·dm^–3^ of P; 397 mg·dm^–3^ of K; 68 mg·dm^–3^ of Na; 1.51 cmolc·dm^–3^ of Ca; 1.0 cmolc·dm^–3^ of Mg; 5.49 cmolc·dm^–3^ potential acidity; 9.9 cmolc·dm^–3^ cation exchange capacity; and 45.2% base saturation. Liming was performed two months before planting to increase base saturation to 70%, and 1.0 kg·m^–2^ of chicken manure tanned in the planting fertilization was also used.

Chicken manure was stored for six months and, at the time of its application, presented the following characteristics: pH 7.7 in water; electrical conductivity of 26.3 dS·m^−1^; water retention capacity 2.2 mL·g^−1^; apparent density 0.4 g·cm^–3^; total carbon 41.2%; organic matter 81%; total nitrogen (N) 4.5%; N-ammonium 358.0 mg·kg^−1^; N-nitrate 31.9 mg·kg^−1^; total phosphorus (P) 8.6 g·kg^−1^; total potassium (K) 36.8 g·kg^−1^; sodium (Na) 4.4 g·kg^−1^; calcium (Ca) 31.1 g·kg^−1^; (1 g); magnesium (Mg) 11.7 g·kg^−1^; sulfur (S) 6.2 g·kg^−1^; boron (B) 46.6 mg·kg^−1^; manganese (Mn) 699.0 mg·kg^−1^; and zinc (Zn) 644.0 mg·kg^−1^. Cover fertilization was performed at 10 and 20 days after transplanting the seedlings. Urea 0.1% was used in foliar fertilization in irrigation water. In total, 380, 90, and 280 kg·ha^−1^ of N, P_2_O_5_, and K_2_O were administered to plants according to plant requirements and phenological stages.

The field trial was established in a randomized block design, with four repetitions, and each plot was composed of ten valuable plants of the exact clone. The spacing used was 3 × 1.0 m between plants, and the holes were set with dimensions of 40 × 40 × 40 cm. The test comprised 600 useful plants and two rows of borders in each block. The seedlings were planted on 31 January 2019, when they had between 4 and 6 pairs of fully expanded leaves. The management and development of culture and the determination of cultural tracts were carried out according to the necessities and the technical recommendations for the culture [5].

Fifteen clones propagated by cuttings were studied: ten Conilon x Robusta hybrids from the *C. canephora* Breeding Program of Embrapa Rondônia and five clones from the cultivar Conilon—BRS Ouro Preto (Table 3).

### 4.2. Collection and Preparation of Samples from Soil, Roots, Leaves and Fruits

In the flowering period, samples were collected from the leaves of the plagiotropic branch and the middle third of the roots and rhizospheric soils of each plant of the previously identified plot for multielemental analysis. For leaf analysis, samples were collected from all plants of the three blocks. Roots and rhizospheric soils were collected only from the plants of the first block due to the large volume of analyses required in the TXRF. The depth of the collection of rhizospheric soils and roots was 0 to 20 cm, from openings held near the neck of the plant with the aid of a Dutch-type auger.

The preliminary preparation of samples from rhizospheric soils, roots, leaves, and grains of *C. canephora* clones was carried out in the plant breeding laboratory of the UFAM. The samples of rhizospheric soils were dried at room temperature and sieved in 4 mm mesh. According to the Brazilian Soil Classification System, the subsamples were passed through 2 mm meshes to obtain the air-dried fine earth [68]. The determination of its pH was performed according to the methodology of Boyle [69].

The samples of leaves and roots were washed with a water solution plus neutral detergent (1 mL·L^−1^), running water, distilled water, and deionized water. The roots were washed in a sieve with a mesh of 1 mm in running water for one minute and then subjected to triple washing with distilled water and later with ultrapure water. After the washing process, the samples were dried in an oven with forced circulation of air at a temperature of 65 °C until they reached constant weight. The dry mass of leaves and roots and total dry mass in grams were obtained using a Mettler PM 30-K scale (Mettler Toledo, Columbus, OH, USA) according to the protocol established by Dickson et al. [70].

The clones were harvested when 80% of the fruits were in the red-cherry stage. Samples were collected separately for each clone that went through the precleaning process and dried naturally according to the method by Ferrão et al. [4]. The grain yield (60 kg bag) and the productivity of the processed coffee (60 kg bags·ha^−1^) of the clones were obtained according to the method of Moraes et al. [7].

A sample of peeled coffee beans from each plant of the first block was placed separately in a “WEREW” electric roaster, at a temperature of 200 °C, for 20 min until reaching the roasting point with an average dark overall appearance, according to roasting and flavor recommendations based on Brazilian preference [71] for the comparative analysis of raw grains harvested from the same plant.

For TXRF analysis, the dried samples of the rhizosphere soils and the different plant tissues were ground manually and stored in microtubes of 2.0 mL previously identified and separated, corresponding to each clone. Subsequently, they were pulverized in a vibratory grinder (model MM400/Retsch GmbH, Haan, Germany). Then, the samples were weighed, leaving 50 mg of the material of rhizospheric soils, roots, leaves, and raw and roasted grains for further analysis. After weighing, 1.5 mL of a solution (e.g., 1% aqueous Triton X100) was added to each microtube to be suspended and homogenized in a vortex. After homogenizing the samples, 10 µL of an internal standard of Gallium (Ga) was added [72].

### 4.3. Analysis Using the Total Reflection X-ray Fluorescence Analytical Method

The TXRF analysis was performing at the scientific–technical laboratory of chemical analysis of the Regional Superintendence of the Federal Police of Amazonas. For the reading of the samples, a bench spectrometer, model S4 T-STAR/Bruker (Bruker, Billerica, MA, USA), equipped with two X-ray tubes (an anode X-ray tube Molybdenum Mo at 17.5 keV and a cathode consisting of a tungsten filament (W)) was used to facilitate excitation at 35 keV [72].

The samples were pipetted onto non-siliconized and dry quartz supports, which were previously prepared with 10 µL of polyvinyl chloride (PVC) pipetted in the center. They were dried on a heating plate at approximately 100 °C. The discs were dried again after 10 µL of the sample suspension was pipetted and allocated to the columns corresponding to the analysis map injected into TXRF [73]. The blank samples and duplicates of rhizospheric soils and *C. canephora* tissues were analyzed.

The measurement time of the samples was determined to be around 600 s to generate quality in the spectrum. Quartz carriers were applied as sample holders and reflectors. For the determination of the relative sensitivities of the elements, gallium (Ga) was used as a reference element due to its known concentrations, followed by a sensitivity calculation [74], which is based on the peak area of each element as follows:(1)$$S_{i}=\frac{N_{i}\cdot C_{Ga}}{N_{Ga}\cdot C_{i}}$$
where
*N_i_*: net peak counts of a given element;*N_Ga_*: net counts of Ga peak;*C_i_*: concentration of a specific element in the solution;*C_Ga_*: concentration of the element Ga in the solution.

Using this methodology, and considering the unit of measurement as mg·kg^−1^, the following elements were detected: phosphorus (P), sulfur (S), chlorine (Cl), potassium (K), calcium (Ca), titanium (Ti), vanadium (V), chromium (Cr), iron (Fe), cobalt (Co), nickel (Ni), copper (Cu), zinc (Zn), bromine (Br), rubidium (Rb), strontium (Sr), yttrium (Y), hafnium (Hf), and iodine (I).

### 4.4. Statistical Analysis

The results of grain yield and productivity and chemical element concentrations in the leaves were submitted for analysis of variance and compared using the *F* test (*p* < 0.05); the means were grouped using the Scott–Knott test (*p* < 0.05). Experimental coefficient of variation (CVe), coefficient of genetic variation (CVg), coefficient of genotypic determination (CVg/CVe), and heritability (H^2^) were estimated [75].

Multivariate analyses were performed for concentrations of chemical elements in roots, leaves, and raw and roasted grains of clones, as well as for grain yield and productivity. The results were grouped according to the hierarchical method of means of distances (UGPMA—unweighted pair-group method using an arithmetic average) using the mean Euclidean distance matrix from the program R (RStudio Team, Boston, MA, USA) [76] and its complement RStudio Team (RStudio Team, Boston, MA, USA) [77]. Subsequently, the non-metric multidimensional scaling (NMDS) analysis was carried out, in which the more excellent dispersion of the units was identified as having more significant genetic dissimilarity. The Vegan package of the R program was used to perform the analyses [78].

The procedure started from an initial organization that interactively reorganized individuals to reduce stress (standard residual sum of squares). Stress (S) is a function that indicates the magnitude of the loss of information in the dissimilarity matrix using the procedure, which measures how the positions of individuals in an n-dimensional configuration deviate from accurate distances (dissimilarities) after scaling. Stress was determined using the value of R^2^ = 1 − S^2^ [78]. The study of the chemical element’s relative importance in the genetic diversity prediction was also performed according to the methodology proposed by Singh [22] for the variables of leaf chemical elements, grain yield, and productivity, and it was designed with replications. The Pearson correlation analysis between the characteristics studied was performed with the Genes program (Viçosa, MG, Brazil) [75]. The dry mass of the leaves and roots and total dry mass were used to study the correlations of the characteristics.

## 5. Conclusions

The phenotypic characterization for chemical elements in controlled coffee experiments using TXRF allows for the discrimination of genotypes to identify the most efficient absorption of macronutrients and essential micronutrients, as well as potentially toxic trace elements, in the plants. The CVg/CVe ratio and high heritability for the concentration of Br and Hf in leaves and grain yield show more favorable conditions for selection in plant breeding in the discrimination of genotypes.

*Coffea canephora* absorbs and accumulates the chemical elements Rb, Ti, Y, Sr, and Hf in tissues and grains; this is not yet reported for the species, and the concentrations and effects of these elements on plant, human, and animal health, as well as on interaction with other elements, are not yet known.

TXRF is an efficient technique to characterize chemical elements’ presence and concentration values, expanding the possibility of establishing breeding programs to discriminate fast-growing genotypes for plant breeding in *C. canephora.*

## Figures and Tables

**Figure 1 plants-12-04052-f001:**
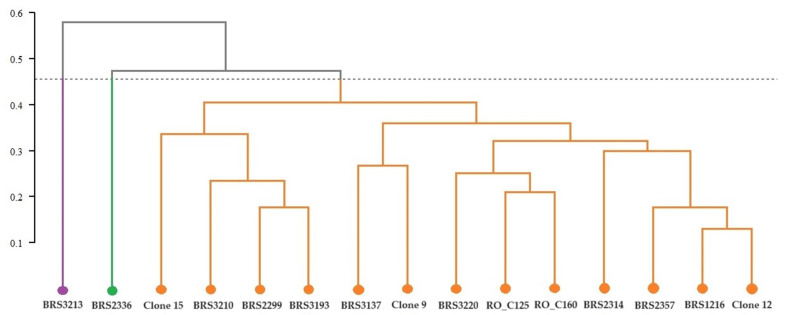
Cluster analysis (UPGMA) from the average Euclidean distance between 15 genotypes of *C. canephora* based on analysis of chemical elements present in the roots of the plants. Cophenetic correlation coefficient: r = 0.8835. The dashed horizontal line represents the cut-off estimated using the Mojena method [21]. All clones that share the same color are part of the same group.

**Figure 2 plants-12-04052-f002:**
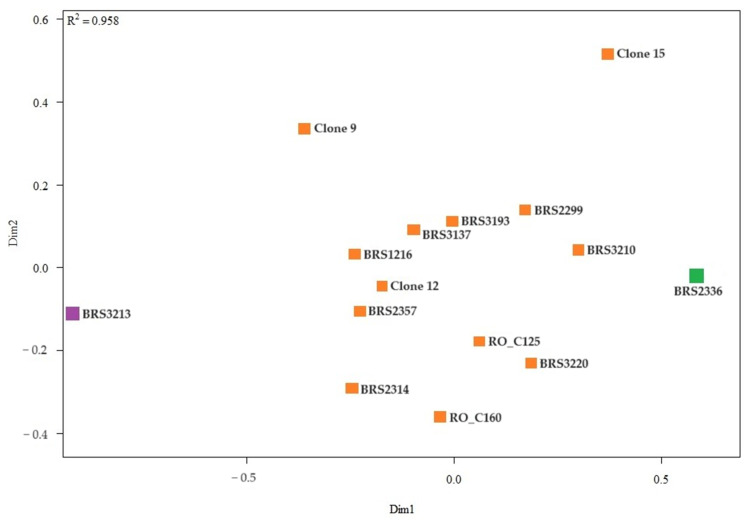
Analysis of genetic dissimilarity of *C. canephora* clones based on root samples. All clones that share the same color are part of the same group.

**Figure 3 plants-12-04052-f003:**
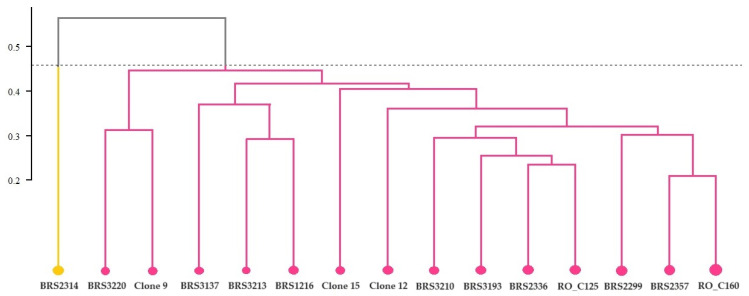
Cluster analysis (UPGMA) from the average Euclidean distance between 15 genotypes of *C. canephora* based on analysis of chemical elements present in the leaves of the plants. Cophenetic correlation coefficient: r = 0.8835. The dashed horizontal line represents the cut-off estimated using the Mojena method [21]. All clones that share the same color are part of the same group.

**Figure 4 plants-12-04052-f004:**
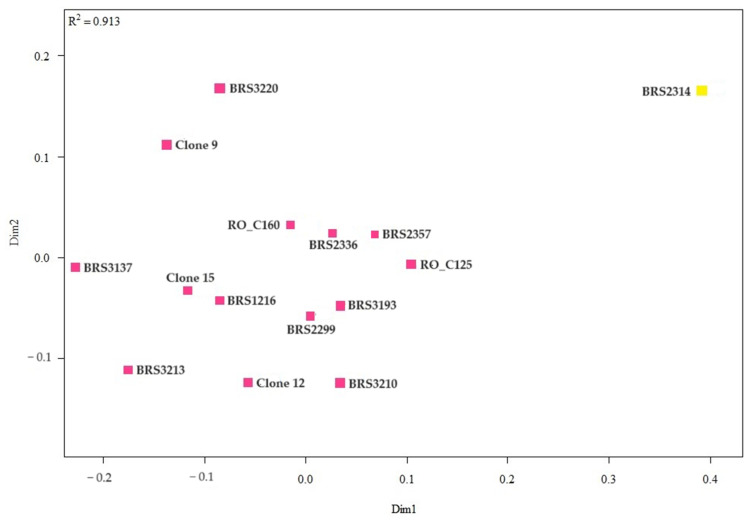
Analysis of genetic dissimilarity of *C. canephora* clones based on leaf samples. All clones that share the same color are part of the same group.

**Figure 5 plants-12-04052-f005:**
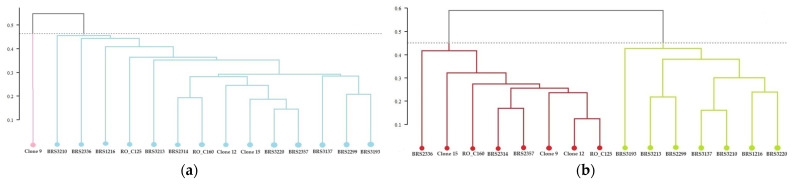
Cluster analysis (UPGMA) from the average Euclidean distance between 15 genotypes of *C. canephora* based on analysis of chemical elements present in the raw (**a**) and roasted (**b**) grains of *C. canephora*. Cophenetic correlation coefficient: r = 0.8835. The dashed horizontal line represents the cut-off estimated using the Mojena method [21]. All clones that share the same color are part of the same group.

**Figure 6 plants-12-04052-f006:**
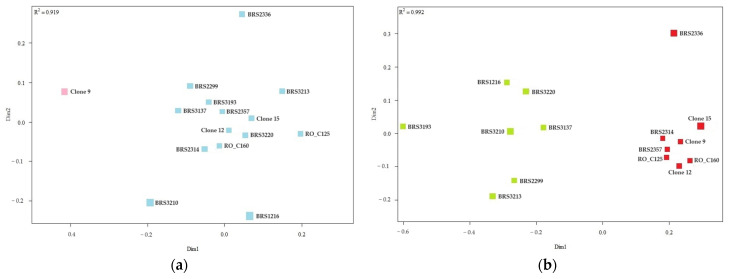
Dissimilarity analysis of *C. canephora* clones based on raw (**a**) and roasted (**b**) grain samples. All clones that share the same color are part of the same group.

**Figure 7 plants-12-04052-f007:**
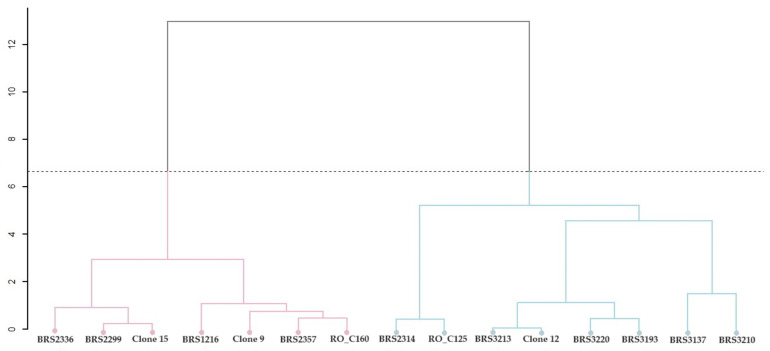
Cluster analysis (UPGMA) from the average Euclidean distance between 15 genotypes of *C. canephora* based on analysis of grain yield and productivity. Cophenetic correlation coefficient: r = 0.8835. The dashed horizontal line represents the cut-off estimated using the Mojena method [21]. All clones that share the same color are part of the same group.

**Figure 8 plants-12-04052-f008:**
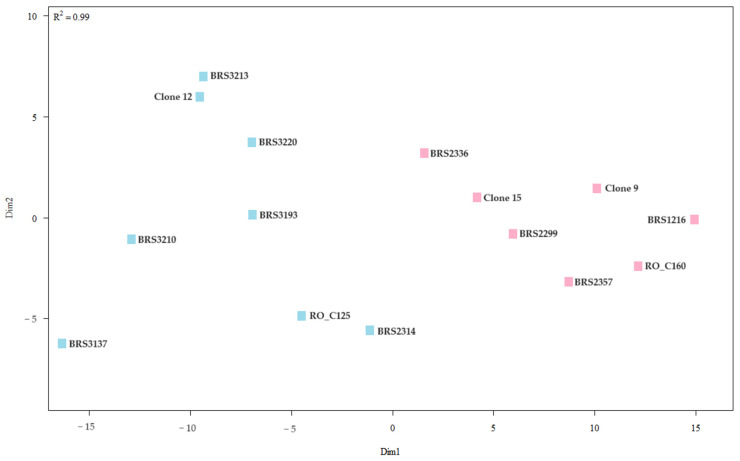
Analysis of genetic dissimilarity of *C. canephora* clones based on grain yield and productivity. All clones that share the same color are part of the same group.

**Figure 9 plants-12-04052-f009:**
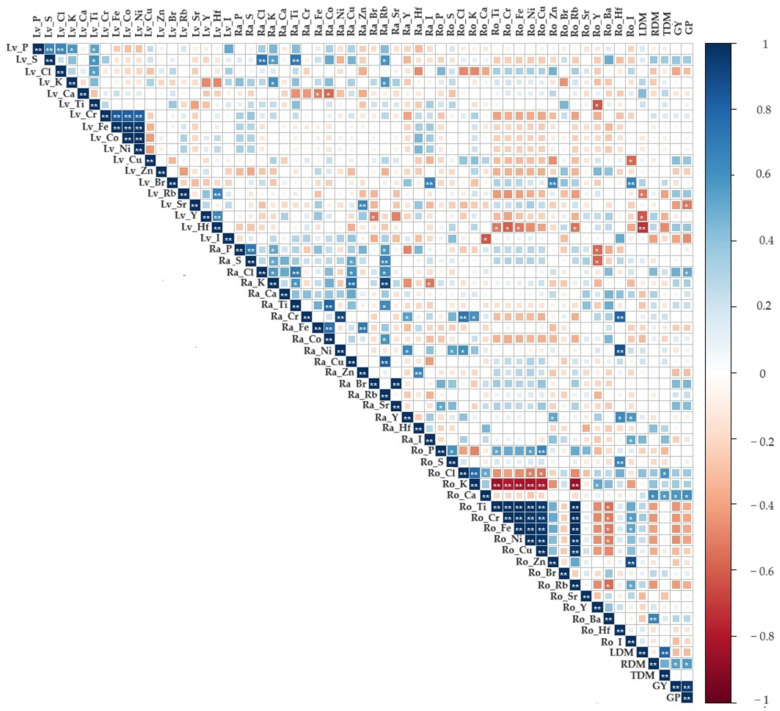
Pearson’s correlation between inorganic components for leaves, biomass, and raw and roasted grains of 15 *C. canephora* clones: Lv (leaves), Ra (raw grains), Ro (roasted grains), LDM (leaf dry mass), RDM (root dry mass), TDM (total dry mass), GY (grain yield), and GP (grain productivity). Colors refer to negative (red) and positive (blue) correlations, respectively—the more intense the color, the higher the correlation. Asterisks indicate significance level (* *p* ≤ 0.05 and ** *p* ≤ 0.01).

**Table 1 plants-12-04052-t001:** Estimates of the experimental coefficient of variation (CVe), coefficient of genetic variation (CVg), coefficient of genotypic determination (CVg/CVe), heritability (H^2^), general average, and standard deviation for concentration in mg·kg^–1^ of elements in the leaves and for grain yield (60 kg bag) and productivity (60 kg bags·ha^–1^) of 15 clones of *C. canephora*.

Characteristics	*F*-Value	CVe (%) ^1^	CVg (%) ^2^	CVg/CVe (%) ^3^	H^2^ (%) ^4^	General Average and Standard Deviation *
Phosphorus	0.0561 ^ns^	6.01	1.74	0.29	25.05	2944.53 ± 1413.00
Sulfur	0.0000 ^ns^	14.00	-	-	-	2824.82 ± 2237.00
Chlorine	0.0803 ^ns^	9.30	-	-	-	2237.53 ± 1580.00
Potassium	0.0603 ^ns^	5.32	1.26	0.24	18.19	17,451.75 ± 9240.00
Calcium	34,149,841.83 ^ns^	16.11	-	-	-	12,719.93 ± 9403.00
Titanium	0.2734 **	44.04	38.39	0.87	75.25	2002.73 ± 2633.00
Vanadium	0.00169 ^ns^	64.02	-	-	-	0.19 ± 0.01
Chrome	0.0632 ^ns^	37.88	4.20	0.11	4.69	78.24 ± 155.00
Iron	1615.25 *	36.39	20.84	0.57	56.75	84.63 ± 49.00
Cobalt	0.0238 ^ns^	67.75	29.42	0.43	43.00	1.00 ± 1.00
Nickel	0.2694 ^ns^	40.54	11.38	0.28	23.97	167.86 ± 297.00
Copper	63.6569 ^ns^	45.95	3.18	0.07	1.88	354.15 ± 502.00
Zinc	0.668 **	13.43	8.87	0.66	63.59	1423.99 ± 2457.00
Bromine	0.4484 **	30.56	35.68	1.17	84.50	1070.60 ± 1440.00
Rubidium	133.5920 *	52.19	29.26	0.56	55.70	2905.17 ± 2115.00
Strontium	3.0973 **	17.46	15.89	0.91	76.81	402.81 ± 736.00
Yttrium	3.4060 **	30.67	25.38	0.83	73.26	2034.42 ± 1498.00
Hafnium	7066.8230 **	41.31	76.37	1.85	93.19	2321.92 ± 1326.00
Iodine	4093.8764 ^ns^	181.11	50.34	0.28	23.61	161.08 ± 135.00
Grain yield	0.0044 **	14.30	10.35	0.72	67.69	0.26 ± 0.03
Productivity	3896.3523 **	25.95	43.75	1.69	91.91	68.39 ± 15.56

^1^ Experimental coefficients of variation; ^2^ coefficient of genotypic determination; ^3^ coefficient of genotypic determination; ^4^ heritability; * Appendix A presents the values of the mean and standard deviation for the clones. ^ns^, **, and *: not significant, significant at 1% and 5% probability, respectively, using the *F* test.

**Table 2 plants-12-04052-t002:** The relative contribution of chemical elements present in the leaves to divergence among 15 *C. canephora* genotypes, using the statistic (S.j) proposed by Singh [22].

Element	S.j	Relative Contribution (%)
Hafnium	593.5935	16.35
Rubidium	511.4090	14.09
Cobalt	436.6348	12.03
Iodine	290.8561	8.01
Copper	257.7923	7.10
Potassium	239.9152	6.61
Chrome	157.8306	4.35
Calcium	155.5942	4.29
Iron	139.0252	3.83
Yttrium	130.9612	3.61
Nickel	130.4908	3.59
Bromine	125.1081	3.45
Strontium	116.5392	3.21
Sulfur	110.4130	3.04
Titanium	98.4062	2.71
Vanadium	51.8913	1.43
Chlorine	37.6019	1.04
Phosphorus	30.4774	0.84
Zinc	15.4727	0.43
Total	3630.0100	100.00

**Table 3 plants-12-04052-t003:** *Coffea canephora* clones evaluated.

Clone	Origin
BRS3137	Open pollination
BRS3213	Encapa03 × Robusta2258
BRS1216	Encapa03 × Robusta1675
BRS2314	Encapa03 × Robusta640
BRS3210	Encapa03 × Robusta2258
BRS3220	Encapa03 × Robusta1675
BRS2336	Open pollination
BRS2299	Open pollination ^1^
BRS3193	Open pollination
Clone 12	Conilon—BRS Ouro Preto
BRS2357	Open pollination ^1^
RO_C125	Conilon—BRS Ouro Preto
RO_C160	Conilon—BRS Ouro Preto
Clone 9	Conilon—BRS Ouro Preto
Clone 15	Conilon—BRS Ouro Preto

^1^ Open-pollination genotypes from the cultivar Conilon—BRS Ouro Preto, developed by EMBRAPA, in 2013. Encapa: Conilon.

## Data Availability

Data are contained within the article and Appendix A.

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
