# Peer review of "Phenotypic Diversity and Genetic Parameters of Coffea canephora Clones"

_plants, 2023, doi:10.3390/plants12234052_

Round 1
Reviewer 1 Report
Comments and Suggestions for Authors
The article's innovation lies in its development of a novel technique, TXRF, which broadens the scope of chemical element studies, thereby holding significant importance for the assessment of grain composition. This research extends the frontiers of knowledge and contributes valuable data regarding previously unexplored elements absorbed and accumulated in plant tissues and grains: Rubidium, Titanium, Yttrium, Strontium, and Hafnium. I suggest the following changes on relevant points:
- I suggest adding the word heritability as a keyword to contemplate and cover all content and analytics.
- In the introduction and wherever else in the text, change the word plant organ to plant organ tissues.
- In the methodology, the authors report that rhizospheric soils were analyzed for each clone. In the supplementary figure 1, the authors present the result analysis of rhizospheric soils. It should be made in correspondence and put rhizosphere soils and not just soils.
- The authors in the introduction of the article only talked about the advantages of the TXRF technique but did not report the disadvantages. They can report in the discussion, for example, that for results with the methodology used to detect the maximum of elements, they did not detect nitrogen which is an essential macronutrient. And justify that it is of the nature of the technique, but they must present, even if the focus of the article is not on macronutrients, that it is for the maximum of chemical elements. The reader should be aware that it will not detect nitrogen and that other complementary analyses, if it is to focus on nutrition, should be performed.
- In the Supplementary Figure in the title, there is a species name without italics. This format needs to be revised carefully. In the title of the supplementary table, the scientific name is also not in italics.
Comments on the Quality of English Language
I suggest a review of the following points:
- Abstract, in the line "27" change "determined " for " quantified ".
- Introduction, in the lines “41” change "worldwide, and between October 2021 and September 2022, considering" for " worldwide. Between October 2021 and September 2022, considering"- In line "43" change “worldwide, respectively. " for "globally, respectively ".
- In the line "46" change “region" for "regions".
- In line "46" change "This has become an important agricultural activity for the states of Amazonas and Rondônia, as well as for states in other regions such as Espírito Santo (Southeast) and Bahia (Northeast) " to " This has become an significant and lucrative agricultural activity benefitting the states of Amazonas and Rondônia, as well as states in other regions such as Espírito Santo (Southeast) and Bahia (Northeast) ".
- Results, in line "145" change "BRS1216 e Clone 12 " to "BRS1216 and Clone 12".
- In all figures about cluster analysis (UPGMA) change " the method of Mojena " for " the Mojena method".
- In line 586 change “Rubidium, Titanium, Yttrium, Strontium, and Hafnium” for their symbols.
- In line "596, 597 and 598" the sentence must be presented in English.
- Complete the English revision for the entire text with the support of a native speaker.
Reviewer 2 Report
Comments and Suggestions for Authors
In a critical review of this research article, several aspects stand out as both strengths and potential areas for improvement. The study's focus on the simultaneous analysis of a wide range of chemical elements within Coffea canephora clones is commendable, as it offers a more comprehensive understanding of the plant's chemical composition, which is essential for breeding programs. The use of total reflection X-ray fluorescence (TXRF) is an innovative and efficient method for characterizing multiple elements, enhancing the precision and depth of analysis.
However, there are some critical considerations to take into account. The review might benefit from more detailed discussion on the practical implications of the findings, particularly regarding how this increased understanding of chemical elements in Coffea canephora clones can be leveraged in breeding programs. Additionally, the article could elaborate on the limitations and potential sources of error associated with the TXRF method to provide a more well-rounded assessment of its utility.
Moreover, it's essential to address the broader significance of the research within the field of plant breeding and agriculture, such as its potential impact on crop yield, quality, and environmental sustainability. Furthermore, it could discuss the generalizability of these findings to other plant species and regions, as well as the feasibility of implementing these methods on a larger scale. Some specific comments are given below
Before objectives, authors must clearly elaborate on research gaps/hypotheses and the novelty of the study in the introduction section. The objectives may be improved to be more specific
The discussion section may be shortened and authors must not repeat the results in this section, rather they are encouraged to justify the results based on recently published literature
The conclusion must be concrete. It should not be the summary of the results. Authors are encouraged to develop the conclusion based on objectives, give strengths and weaknesses of the study and prospects of the study
Comments on the Quality of English Language
It is Fine
Reviewer 3 Report
Comments and Suggestions for Authors
This study reported a substantial amount of effort on assessment of productivity, grain yield and concentration of chemical elements. The result is useful to improve the understanding of phenotypic diversity in Robusta coffee.
I have several minor comments on the clarity and coherence of this manuscript. These comments are mainly related to the results interpretation and discussion of this paper.
1. The title should be more precise and accurate to reflect the elements of “productivity, grain yield and concentration of chemical elements”, because the study is not about general phenotypic characteristics on morphological or agronomic traits.
2. It appeared that some traits have significant clone effect whereas the others don’t. It would be helpful to have a summary ANOVA table showing the difference. This ANOVA table can be either included in the main text or as a supplemental data.
3. It’s well known that yield and concentration of chemical elements are significantly affected by environmental factors. The result of this study is based on a single location. While the result is still valid, the authors need to acknowledge that results from different environments will be necessary to estimate genetic parameters (to separate variation caused by Environments and G x E). I understand that it’s difficult to run multi-location trial for coffee. However, data generated from multi-year observation will be helpful.
4. Same as point 2, one of the highest heritability estimations was found in productivity (>90%). This is much higher than reported heritability of productivity in previous literature. It’s well known that productivity typically has low heritability. Previous literature need to be reviewed and additional discussions on this observation is needed.
5. The experiment used 15 clones. Among them there is one group that are half-siblings (progeny of ‘Encapa03’). However, results of cluster analysis (UPGMA and MDS) did not reflect their relationship as siblings. Explanation is needed for these disparities.
Comments on the Quality of English Language
Please see the comments to authors for details.
Round 2
Reviewer 2 Report
Comments and Suggestions for Authors
The authors have significantly improved the manuscript and it can be processed for the publication
Comments on the Quality of English Language
Only minor grammatical imporvements required